# The Value of Reentry Housing, Zoning, and "Not in My Back Yard" (NIMBY) Obstacles, and How to Overcome Them

Ivis García 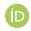

Department of Landscape Architecture & Urban Planning, Texas A&M University, Architecture Center Bldg. C, 3137 TAMU, College Station, TX 77840, USA; ivis.garcia@tamu.edu; Tel.: +1-(979)-393-1293

**Abstract:** Given the housing issues that people who have been in prison face, this article examines the permitting process to operate a vocational and life skills training program for the formerly incarcerated in Salt Lake City, UT, called The Other Side Academy (TOSA). This article employs participant observation, personal and public meeting conversations, planning division staff reports, public comments, and newspaper articles to answer the following question: How was TOSA described in the public input process for a conditional use permit? The author examines how TOSA neighbors first opposed the project and then came to support it. But even with community support, planning staff struggled to find a zoning code that would allow TOSA operations. In the end, the final decision to approve the conditional use permit came to the hearing officer, who sided in favor of TOSA. In this case, planners can learn about the zoning obstacles that reentry housing faces and how those obstacles can be overcome. Finally, academia and planning divisions need to better educate planners involved in administrative process into understanding the intent of the code to achieve just outcomes.

**Keywords:** formerly incarcerated individuals; group home; recidivism; reentry; zoning





## 1. Introduction

Given the growth of the U.S. population in federal/state prisons and local jails since the early 1970s, from 200,000 to over 2 million, it is expected that more communities need to plan for reentry housing [1–3]. According to the U.S. Department of Justice, about 650,000 prisoners are released each year [4] and usually return to communities without a home or a job, and in need of treatment and vocational education [2,3,5,6]. Thus, it has become more critical than ever that policymakers and communities make residential, treatment, educational, and employment accommodations for those exiting correctional facilities. To reduce the likelihood of recidivism, cities need to be able to support reintegration to society in a holistic manner [2,7–10].

This article is concerned with equity planning and how advocacy, land use, and zoning policy affects the placing of formerly incarcerated individuals living in a group setting when trying to locate in neighborhoods [11–13]. Reentry housing might be zoned under a number of categories from community correctional facilities to halfway houses to group homes. Regardless of their classifications, any home or facility for the formerly incarcerated is likely to encounter "Not In My Back Yard" (NIMBY) opposition from neighbors because of fear of crime [8,14–16]. But, even if they do not face resistance, they might face zoning obstacles which have not been discussed specifically in the planning literature [3,17].

First, the article will review the literature on the topics of (1) incarceration, recidivism, and stigma associated with this understudied population to show the value of reentry housing; (2) the NIMBY opposition, zoning, and legal challenges that these types of projects encounter, as well as how they could be overcome through advocacy and community education. A case study of the approval process for The Other Side Academy (TOSA), a reentry home/vocational program, adds to the literature on the intersection of urban planning, advocacy, and reentry.

Second, the method sections explain the data analyzed—participant observation, reports, and public hearing transcripts—to be able to answer the question: How was TOSA described in the public input process for a conditional use permit? Third, in the findings, section two main topics are discussed: (1) the value of TOSA as described in the public input process; (2) zoning and legal obstacles. Fourth, we discuss the article's empirical contribution by digging deeper into how TOSA has gained support from the public and how administrative processes can result in unjust outcomes. The discussion makes a theoretical argument for why planners need to understand the intent of zoning ordinances and be empowered for equitable outcomes as opposed to hiding in technicalities. Finally, the conclusion will discuss how planning actors can overcome zoning difficulties of NIMBY politics by finding ways of educating those involved in the planning process—from the planning staff to the public.

There are limitations to single case studies, such as non-generatability, the possibility of bias based on the authors standpoint, the dependence on written materials and transcripts, and the absence of a final case evaluation regarding how the input affects use permits. Despite the drawbacks of single case studies, they also serve to tell a compelling story, comprehend the connections between the different actors, and enable an analysis of the decision-making processes and actions of local governments [18].

## 2. Overview of the Current State of the Art in the Literature

### 2.1. Incarceration, Recidivism, and Housing Difficulties

Within only three years, it is estimated that about two-thirds of those released from prison are rearrested, and about 40 percent will return to prison [4]. Among drug users, recidivism is especially high [10,19,20]. Therefore, one of the most pressing questions among academics and policymakers has been how to reduce the probability of rearrests [3,5,6]. Previous research shows that, outside of demographic characteristics such as age, gender, race, ethnicity, and neighborhood socioeconomic status, there are other factors that contribute to the likelihood of recidivation, such as low educational attainment, a history of unemployment, substance abuse disorders, and prior offenses [2,3,21].

The stigma associated with having a criminal record also affects their housing choices and opportunities [3,10,21–23]. The overwhelming majority of formerly incarcerated persons end up living with their families and friends after being released [23,24]. Nonetheless, the chance of returning to prison increases when people come back to the same environment (friends, family, etc.) that resulted in imprisonment in the first place [25].

This is because most formerly incarcerated individuals lived in neighborhoods with high crime rates that did not offered them opportunities before they were incarcerated [22]. For many formerly incarcerated persons, living with family or friends is not an option or a desirable choice [8]. Most turn to the private housing market [26]. Many public housing authorities and landlords require drug tests, background checks, credit scores, and proof of employment; these requirements represent obstacles for formerly incarcerated individuals [3,8,27]. Furthermore, given that most formerly incarcerated persons will earn low wages, the main barrier against finding housing is merely the ability to find affordable housing; many subsidized housing programs have waiting lists, and many preclude formerly incarcerated individuals from accessing housing altogether [1–3,28].

Although formerly incarcerated persons encounter difficulties finding a home on their own, and only a minority end up living in some supportive housing program where they do not only receive a housing benefit, but they also might receive case management, vocational training, and substance and medical treatment [23,24,26]. Without subsidized or supportive housing, it is estimated that many formerly incarcerated individuals end up homeless [2,26,29]. There is evidence that access to a stable home first is essential to be able to address other issues such as unemployment and substance abuse disorders [21,30,31]. Thus, having a residence not only facilitates successful reentry but also reduces recidivism [10,25,28].

Another strategy that has been demonstrated to reduce recidivism is the mentorship of formerly incarcerated persons through "professional ex-" programs, helper/wounded healer models, and mutual aid groups [9,32–34]. All of the models above take a strength-based approach which concentrates on the assets that the formerly incarcerated already have to achieve community reintegration [6,32,35,36]. Under a strengths-based approach, people learn to advocate for themselves by recognizing that they are of value and they can be beneficial to society [6,37]. Often, clients are employed by the same organization that provides them with housing; they often engage in a program where they practice helping others, contributing to their neighborhoods, and reducing crime by being part of the solution and not the problem [34].

In this way, group homes and vocational programs with peer or mentor support meet the criteria for of serving the public good, which provides a rationale for the role of planning in reentry initiatives [38]. Still, reentry measures might generate conflict and lack public or municipal support, as described in the next section [39].

*2.2. NIMBY Opposition and Zoning Issues Encountered by Group Homes*

Within the planning process, vocational programs and housing for formerly incarcerated individuals can encounter sitting conflicts or "Not In My Back Yard" (NIMBY) opposition [8]. NIMBYs are organized opposition movements against a locally unwanted land use (LULU) [14,16,38,40,41]. Research on NIMBYs has concentrated on (1) the geography, socioeconomic, and political characteristics of communities chosen—they are usually in low-income and minority areas, with transient populations, fewer families with children, and little political clout; (2) when and how residents organized for opposition, and the more significant structural changes that led to the growth of group homes such as de-institutionalization, deindustrialization, and increasing inequalities [14,42,43].

Support or opposition might be related to the kind of human service facility being proposed (i.e., mental health facility, HIV/AIDS, people experiencing homelessness, children, etc.) [16,38,41,43]. Research has shown that educating stakeholders in the siting process and addressing their concerns can result in acceptance, transforming a movement from NIMBY to "In Our Backyard" [8]. Research in the urban planning field has not presented many cases that show how engagement with the residents has the potential to improve housing and vocational programs for the formerly incarcerated. But NIMBY-induced conflicts are not the only barrier to group homes; other kinds of post-prison housing such as land use and zoning regulations also matter greatly.

Legislation such as the Fair Housing Act (1988) has helped to reduce discrimination within planning administrative processes. Still, the planning framework tolerates exclusionary zoning practices where only single-family homes are allowed, excluding not only multifamily homes but also commercial uses [44,45]. In particular, group homes might be rejected because of their commercial nature [8]. Many municipalities, to stay away from potential litigation, might allow group homes but then make some constraints delineated in conditional use permits—e.g., not in particular neighborhoods, restrict distance between group homes, limit the number of occupants, offer additional security measures, require them to obtain approval from neighbors, request crime prevention through environmental design (CPTED) measures, etc. [14,15,17]. Because many cities will require special or conditional use permits, this is where and how planning divisions might impose stringent procedural requirements [17].

Frequently, municipalities might classify these kinds of living arrangements not as group homes but as boarding institutions, clinics, schools, or businesses, to exclude them from residential areas altogether [46]. Because municipalities could classify these institutions in different ways, from group homes to vocational schools, there might be separate administrative avenues which could add to the confusion of which zoning ordinance to apply to [47]. For example, zoning ordinances might allow housing but not commercial activities that, in other circumstances, might be seen as vocational activities. Because of

all of the constraints on group homes, the administrative appeal is inadequate most of the time [47].

Planning scholars have studied the relationship between rationality and political acceptability. Rational planning models might not necessarily result in political acceptability [40,48]. Municipalities have found that residents move from opposition to support after having communication with the group home developers as well as engaging in a meaningful participatory planning process where expert knowledge is provided [8,38,40]. Even though group homes might raise concerns in the planning approval process, previous studies have found that, over time, neighbors are more likely to be supportive than critical of group homes because their interests and concerns have been effectively addressed [8,40].

This means that planning education can influence a favorable political climate in controversial issues [8,49]. When people are informed about issues related to human services—by lectures or interactions with social service professionals or clients—they are more likely to have positive attitudes toward group homes [8,38]. Some studies of group homes have shown, contrary to popular media beliefs, that health services can be excellent neighbors and often improve the neighborhood by increasing safety and even property values [8,14,38].

The kinds of residences and services that are provided to accommodate formerly incarcerated persons affect those individuals' ability to succeed. In particular, supportive housing is associated with successful reentry [10]. Formerly incarcerated individuals depend on group homes with supportive services and personal networks so that they can successfully reintegrate into society [50].

### 2.3. Research Gap

It is important to discuss some of the limitations of the above studies. Firstly, most studies are case studies, focusing on particular populations or regions [3,21,34]. Because many studies have primarily focused on examining the sociopolitical characteristics of communities selected for housing facilities for the formerly incarcerated, they have overlooked zoning factors that may contribute to NIMBY opposition [14,42,43]. This narrow focus limits our understanding of the dynamics involved in NIMBY opposition, as it fails to take into account the specific concerns and motivations of residents. This is why it is crucial to conduct other case studies like the present one to obtain a more comprehensive understanding of this issue, even when is not comparative.

Second, there is a lack of research that delves into the personal experiences and perspectives of individuals who were formerly incarcerated [3,10,21–23]. Exploring their experiences within the cultural contexts they navigate would yield valuable insights into the challenges they face as well as potential solutions.

Third, limited research exists regarding the effectiveness of interventions and policies aimed at reducing recidivism rates and facilitating reentry into society [21,30,31]. It is imperative that we conduct evaluations of reentry programs and initiatives focused on supportive housing to determine their effectiveness while identifying best practices.

Fourth, many studies disregard the structural aspects that contribute to housing challenges. These structural factors include discrimination, insufficient affordable housing options, and limited access to social support networks [9,32–34].

Fifth, it is important to note that existing studies often overlook the role of land use and zoning regulations as obstacles to establishing group homes and other housing initiatives for individuals who have been released from prison [44,45]. These regulations can significantly impact the process of determining locations for housing options thereby impeding their development.

Finally, research, on NIMBYs has mostly concentrated on opposition towards types of human service facilities like health facilities or shelters for people experiencing homelessness [16,38,41,43]. This limited scope neglects the challenges and dynamics associated with housing and vocational programs for individuals who were previously incarcerated. Moreover, although some studies have explored how NIMBY opposition can be transformed

into support through stakeholder engagement and education, there is a lack of evidence demonstrating the effectiveness of these approaches in relation to housing and vocational programs for formerly incarcerated individuals [8]. Further research is necessary to evaluate the effects of engagement strategies and uncover the factors that contribute to shifts, from NIMBY ("Not In My Backyard") to "In Our Backyard" (IOBY) perspectives. Taking these limitations into account, studies like the present one can offer an understanding of NIMBY opposition, stakeholder involvement, and land use regulations within the realm of vocational programs and housing for formerly incarcerated individuals.

## 3. Methods

Investigating real-life examples can be beneficial in recognizing obstacles and possibilities that are specific to a particular community [51]. These may involve challenges like opposition from residents and zoning complications, which might not be fully captured in comprehensive quantitative studies. Delving into the circumstances and dynamics of communities' case studies offer valuable insights into the intricacies and complexities involved in reentry efforts [51]. They also provide guidance on how to overcome these barriers and foster community acceptance.

From a theoretical standpoint, case studies align with the ecological framework which acknowledges the interplay between individual, interpersonal, community, and societal factors in shaping reentry results [52,53]. This framework proposes that achieving reintegration is impacted by elements across different levels, encompassing personal capabilities, community support systems, and the wider social and policy context. Case studies offer a means of evaluating programs and policies and gaining insights into how they contribute to defining processes and outcomes as well as facilitating communication and collaboration [54].

Some of the expected outcomes of this study include the following: (1) understanding the challenges and barriers that reentry housing projects face due to opposition from the community and zoning regulations; (2) investigating how public input processes impact the approval or rejection of reentry housing projects; (3) offering strategies and recommendations to overcome zoning challenges and gain support from the community for reentry housing initiatives; (4) gaining an understanding of the roles played by stakeholders in the planning and approval process of reentry housing projects within a community; (5) contributing insights and lessons learned to the existing body of literature on reentry housing and its acceptance in communities, aiding initiatives.

To conduct the TOSA case study, the tested iterative process involved the following steps: (1) establish research objectives; (2) define the research question; (3) review the existing literature; (4) design the methodology; (5) gather data; (6) analyze data; (7) draw conclusions and write results. Figure 1 shows the iterative tested process where these "steps" are not strictly linear and separate from each other.

The main objectives of this study were as follows: (1) to examine the challenges and obstacles that reentry housing projects face due to opposition from the community and zoning regulations; (2) to investigate how public input processes influence the approval or rejection of reentry housing projects; (3) to identify strategies that could help overcome zoning difficulties and gain community support for reentry housing projects.

To answer the question "How was TOSA described in the public input process for a conditional use permit?", an exploratory case study of municipal planning practice in Salt Lake City in the approval process of a group home and vocational facility for formerly incarcerated individuals was undertaken. The study is based on participant observations of the author, engaging as a planning commissioner of Salt Lake City [55].

Figure 2 shows the different sources of data that were used to conduct the TOSA case study. This included documents like the staff report, which offered insights into the project and its considerations. The staff report, created by the city for TOSA's conditional use permit, was analyzed for content analysis [56]. The public report includes all information that is relevant to be able to make an administrative decision, including attachments—site

plan drawings, elevations, emails, statements, and letters from the applicant, administrative interpretations, and public comments showing support or disapproval. I examined attachments, such as maps, photographs, site plans, and building drawings, to visually depict the property and its surroundings. To enhance my understanding of the project, we incorporated information and existing condition reports. I also considered regulatory perspectives from documents like the Attorneys Office Memo and State Requirements for Recovery Residences. I sought community perspectives by reviewing public process records and comment records. These valuable resources provided insight into sentiment and concerns surrounding the case.

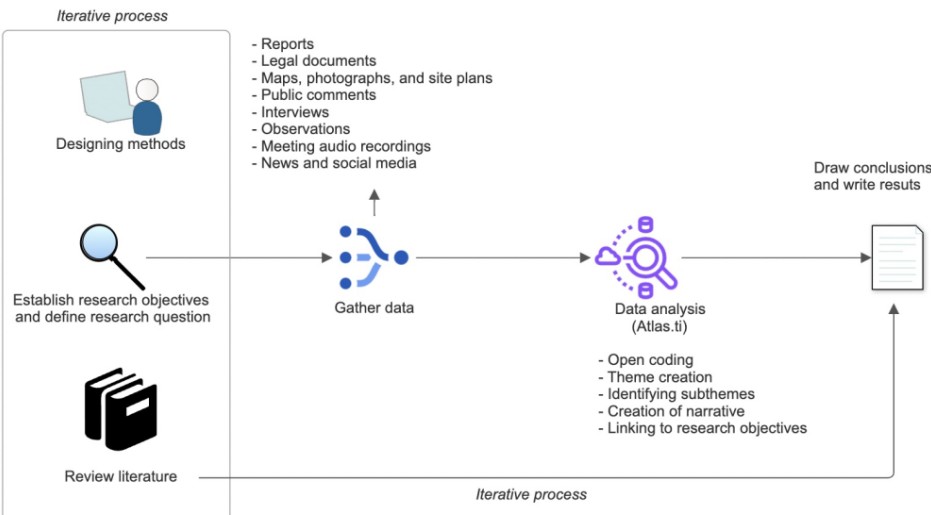

**Figure 1.** Tested iterative process.

1.      Staff Report (Echeverria 2016) (8 pages).
2.      Attachments to Staff Report:
2.1.    Vicinity and Zoning Map (6 pages);
2.2.    Photographs (17 pages);
2.3.    Site Plans and Building Drawings (11 pages);
2.4.    Additional Applicant Information (84 pages);
2.5.    Existing Conditions (2 pages);
2.6.    Analysis of Standards—Conditional Use (11 pages);
2.7.    Attorney's Office Memo (7 pages);
2.8.    State Requirements for Recovery Residences (7 pages);
2.9.    Public Process & Comments (30 pages);
2.10.   Department Comments (3 pages);
2.11.   Property-Related Crime Reports (4 pages).
3.      Meeting Minutes (Planning Comission Meeting Minutes 2017) (9 pages).
4.      Audio Recording (SLCTV 2017) (4 h).
5.      Author's Transcription of 22 March 2017, Meeting Recording (101 pages).
6.      News articles and TV coverage:
6.1.    (Stilson 2017) (2 pages);
6.2.    (Lakhov 2017) (7 pages);
6.3.    (KSLTV 2019) (3:36 min);
6.4.    (KSTU 2018) (1:32 min);
6.5.    (Todd 2019) (2 pages).
7.      TOSA Case Study (Philantrophy Round Table 2017) (17 pages).
8.      TOSA Website (TOSA 2020).
9.      TOSA Facebook (Facebook 2020).
10.     Informal Conversation Fieldnotes (12 pages).

**Figure 2.** Data Sources.

Meeting minutes from The Planning Commission meeting summarized discussions and decisions made during this process. The audio of the meeting, available to the public and posted online, was transcribed verbatim for analysis. Apart from official government documents, media sources played a role in capturing discourse and media representation of the case. I referred to news articles and TV coverage for this purpose. Furthermore, I explored TOSA's website and Facebook page to gain an understanding of their goals and activities. Additionally, valuable insights and contextual information were obtained through conversations and fieldnotes, offering an understanding of personal experiences and different viewpoints. As a planning commissioner, I had informal conversations with the main architect of the project, who also helped me to craft the arguments for this article and provided feedback, etc. I also use fieldnotes from traveling to the site, meetings, etc., which were also analyzed as part of the study. By incorporating an array of data sources, including perspectives and public opinions, the present case study strives to present a thorough comprehension of TOSA project.

The report, the transcript, and other documents produced by TOSA or the city which were related to the case study and fieldnotes were uploaded into Atlas.ti. Thematic analysis methodology was employed, using an inductive approach [57]. First, key direct text lines from written text and quotes from the audio file were identified using open coding [58]. Second, quotes and text-lines were combined into two themes "zoning barriers to housing innovation" and "the value of TOSA". The value of TOSA had two subthemes "housing that saves life's" and "a good neighbor". Third, a narrative was created to link the content lines and quotes into themes to create a storyline.

There are a few limitations to consider in this study. Firstly, the findings may not be widely applicable as they are based on a single case study. Secondly, the author's perspective as a planning commissioner and their interactions with the project heavily influence the study's results. Thirdly, analyzing written documents and transcripts may not fully capture the intricacies and subtleties of the input process. Lastly, it is worth noting that this study does not evaluate the outcomes or effects of input on the conditional use permit. Although single case studies have many limitations, they are used well in planning to (1) tell a story, (2) understand the relationships between rationality, power, and actor behavior, and (3) analyze decision-making processes and municipal action [18].

## 4. Background on Salt Lake City

Salt Lake City, situated in the U.S., serves as Utah's capital and largest city. According to the U.S (Figure 3). Census in 2022, it had a population of 204,657 residents [59]. The city is part of the Salt Lake Metro area, which boasts over 1.2 million inhabitants. When it comes to ethnic composition, in 2022 Salt Lake City is primarily white, accounting for around 71.3% of the population [59]. Additionally, there is a Latino community comprising roughly 19.9% of residents. Other racial and ethnic groups, such as African Americans (1.2%), Asian Americans (5.4%), and Native Hawaiian and Pacific Islanders (1.7%), represent smaller portions of the population.

In recent years, the affordability of living in Salt Lake City has become a growing concern [60,61]. A study conducted by Wood and Eskic (2022) at the Kem C. Gardner Policy Institute of the University of Utah reported that housing prices saw an increase of 28% in 2021 followed by another 23% in 2022 [62]. In July 2022, the median home price in Salt Lake County has crossed the USD 625,000 mark, even though sales have reached a ten-year low [63]. The median income in 2022 was USD 72,357 [59]. This price–income gap means that 70% of households in Salt Lake County cannot afford a home at the median price. The demand for single-family homes has also resulted in increasing rents—which stood at USD 1500 in 2024 [64].Additionally, despite a recent decline in the homeless population, there has been a growth of 14% in 2021 [31,65,66].

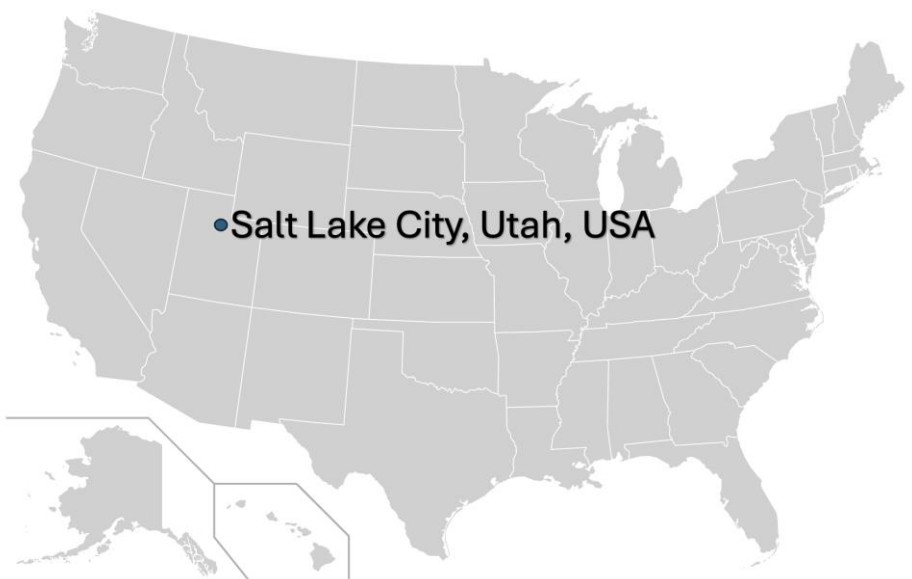

**Figure 3.** Salt Lake City Map Locator. Source: Image created by the author.

To address the issue of homelessness, local government and community organizations have implemented measures such as providing homeless resource centers [31,65,66]. Salt Lake City is mostly known for its landscape, where Republicans hold significant influence. However, in recent years, there has been an increase in political diversity within the city and surrounding county [67]. This shift has resulted in a more varied political climate and increased participation from individuals with different political perspectives. When it comes to incarceration rates, Utah—including Salt Lake City—experiences a lower level compared to the average, that is 176 per 100,000, while the average in the U.S. is 355 per 100,000 [68].

A case study focusing on a scenario like TOSA in Salt Lake City can offer insights into unique contexts and phenomena that may not be easily observed or studied through other research methods. Salt Lake City has encountered hurdles in meeting the housing needs due to its population growth. The combination of a lack of housing affordability alongside their efforts to create more density to accommodate for new housing make it an intriguing case study for exploring the barriers and strategies associated with reentry housing projects.

## 5. Results: The Value of TOSA

The qualitative data gathered provide an understanding of the impact that TOSA has had on both individuals and the community. It is important to note that these results are based on the specific case study of TOSA and may not be generalizable to all reentry housing projects. However, they do offer valuable insights into the potential benefits and challenges of implementing such projects and provide recommendations for overcoming zoning challenges and gaining community support.

The finding focused on quotes from transcriptions of comments, testimonies, and planning commission meetings. Through this analysis, key themes and patterns related to challenges, barriers, and the value of reentry housing projects were identified. The impact of zoning regulations and community opposition was also explored. Additionally, insights from documents, regulatory perspectives, media sources, and stakeholder conversations were incorporated to enhance our understanding of the case study.

The value of TOSA a reentry housing project was highlighted through public comments and testimonies. Attendees spoke about how TOSA saves lives by providing housing, educational, and employment opportunities, as well as cultivating life skills. The vocational training program offered by TOSA was seen as an alternative to clinical treatment, with students gaining skills through hands-on work with affiliated businesses.

The study also examined the zoning obstacles faced by TOSA. The project initially applied to operate as a "congregate care facility" but faced changes in municipal code that eliminated this designation. The planning division later classified TOSA as a "community correctional facility", but the project appealed against this classification. The city lawyer determined that TOSA should be classified as a "recovery residence" instead. The project faced further challenges related to zoning requirements, particularly regarding commercial activities.

Finally, the planning commission debated the inclusion of commercial uses in a group home and eventually approved the conditional use permit, allowing TOSA to operate as a group home with certain conditions. Overall, the findings below highlight the value of TOSA and the obstacles it faced due to zoning regulations and community opposition.

*5.1. Housing That Saves Lives*

Attendees spoke on behalf of TOSA, which was requesting conditional use approval for a multi-site group home for formerly incarcerated people with a history of substance abuse disorders. Most of the public comments, however, were not about the conditional use permit and zoning changes, but about how TOSA reduces recidivism and, ultimately, "saves lives". While in tears, a woman offered a mother's perspective:

> My son was a heroin addict once he got out, he did not have any skills: coping skills, life skills, job skills. His probation officer told him about the academy seven months ago. Now, he is excited to get to work. He is excited! This is a kid that would have killed himself before getting a job. For the first time in 10 years, I hear hope in his voice, so please allow TOSA to do their job: saving lives.

In the same line of thought, a current student declared, "We sell life-changing funnel cakes!" TOSA distinguishes itself from traditional substance abuse disorder recovery residences because it does not provide "clinical treatment". Instead, TOSA runs a "vocational training program" that is affiliated with several businesses, including a moving company, a food truck, a landscaping service company, and an auto detailing service. The program is 95 percent funded by the services that students provide through the vocational school. They receive private donations to cover the remaining costs.

Through a training and residency program, TOSA seeks to address some of the root causes that often result in recidivism and return to substance abuse: the lack of housing, educational, and employment opportunities, as well as uncultivated life skills (responsibility, accountability, and so on). Tara shared her thoughts regarding the need for a holistic approach like TOSA:

> People are talking about zoning, but to put it in perspective, human life is priceless. Stopping to use is not that hard. What is hard is why we use it. It's because there is nothing for us—you have a criminal background, you cannot get a job, and you cannot get housing. TOSA breaks down all those barriers.

Similar to Tara's story, the executive director of TOSA, Dave Durocher, was a drug user for 25 years. He is a graduate of Delancy Street in California, TOSA's inspiration and operational model. Dave was one of the first people who stood up to tell his story:

> I had done a two-year prison term, a five year, six years, and a 10-year prison term. I wrote a letter to Delancey Street, and I went to their Los Angeles family. I did not stay there for two years; I stayed there for eight years. The last five years I manage the LA facility. It literally changed my life, and that is why I am here to do the same thing.

In his role, Dave was accustomed to being a spokesperson for TOSA and telling his story. He has invited residents to visit their site and meet with students and the rest of the staff. Throughout their two-year minimum enrollment period, students must reside on the premises of TOSA. This allows them to develop the skills and behaviors for successful integration into society all within a supportive and career-focused setting. Differently from

other programs, people do not have to pay and can stay at TOSA as long as they want. As one of the managers put it, "until they are ready".

The model is based on a strengths-based model. Formerly incarcerated individuals like Mr. Durocher, with time, become managers. Senior students take leadership roles and manage aspects of the academy. They do not hire professionals because "they will ruin it—causes passivity in students and they will disengage, this is completely based on senior peers", explained a TOSA's board member.

*5.2. A Good Neighbor*

One after another, speakers told stories about how TOSA not only changed lives but how they changed the neighborhood. A man expressed,

> "I've had to explain to my 7-year-old about those houses there, the falling down ones. It's just kind of this corner that we know. So, over the last year and a half I've been noticing these changes taking place".

TOSA has made the area safer by rehabilitating blighted property. TOSA removed from the properties in question many homeless encampments and cleaned up syringes left on the ground for others to find. Their presence in a relatively desolate neighborhood brings eyes to the street and, since they moved in, criminality has reduced according to the staff report [56]. A resident testifies, "now it really is looking like a place with a lot of eyes and ears, to make sure those middle schoolers are a little bit safe in that area. So, as a resident I'm very supportive of what I've seen happening". The TOSA program has night security, where students walk the block. The space is monitored by themselves. The 24 h of active monitoring is to protect their neighborhood and themselves from potentially harmful influences that are part of their past. They also provide free manual labor to neighbors, including shoveling snow and raking leaves. A neighbor describes them as the "finest, hardworking people I have ever met".

Many neighbors spoke about how students always portray a favorable impression of the organization: they always say hello, offer a hand, dress well, and are out in the community. They are indeed an asset to the neighborhood. All the actions above helped TOSA gain acceptance and trust from neighbors. One neighbor said,

> The pictures they have shown is literally what it looks like every day. The trucks are always clean, everybody's shirt is always clean, you'll see groups of spirited guys hanging out, playing basketball. I thought maybe I could take my son down there. I really got interested in just kind of the energy of what was going on.

Another neighbor added, "Every one of these strangers whose home I was invading said hello to me, and many of them stopped to speak to me to make me feel comfortable". Another person expressed, "I've ridden up and back with them twice, and they've become my friends and I call them my boys. Park Lane has entertainment at night for the old people. They sing beautifully. They have come over and sung for us, and I can't say enough about them".

Having a "Good Neighbor Plan" as the chairman of the TOSA's board explained was something that they study from other programs and cities. One of the first things they did as part of the plan was to create "An ongoing committee that will be meeting to address issues that arise in the neighborhood. This is a forum, a format so that you can always be reaching out, working together as a good neighbor within the community to manage the issues, the concerns that arise in the neighborhood", a board member explained.

TOSA always communicated their compelling goals to neighbors of making the community safer and contributed to its stabilization and development while saving lives. They just bought and rehabilitated through self-help (the work of the students) four vacant buildings and lots. Neighbors were particularly interested in TOSA being able to address the dilapidated building on 700 East and the vacant property coined "cocaine row" as one supporter put it. Among the unoccupied properties was the Armstrong Mansion, built in 1893, which was designated as a local historic landmark site and is listed on the National

Register of Historic Places. One neighbor expressed its support by saying, "The work in the historical mansion is truly historical". Students rehabilitated the property themselves, and the changes have been visible to residents. Several of the great-grandchildren of Francis Armstrong, a Salt Lake City two-term Major, showed up to the meeting to raise their concerns. One of them said,

> When I heard that the Armstrong Mansion was going to be a home for ex-convicts, I almost die! I went with my old lady stroll down the street and said: "I need to see what this is all about!" I have never met such charming men and woman—so much nicer than anyone from my street. I see what you are struggling with. You have certain ordinances and regulations. Make the box fit these people because we are going to be mad if you don't. And I'll come strolling down again!

This woman, like others in the neighborhood, admits that she initially showed great resistance towards the reentry housing and vocational program. After she met the students and the staff, having conversations with them about the benefits of supportive housing and the benefits of the development to her neighborhood, she developed trust and support for TOSA. The next section discusses some of the zoning obstacles that TOSA faced.

*5.3. Zoning Obstacles to Housing Innovation*

On 19 August of 2015, TOSA applied for a permit to operate as a "congregate care facility"[1]. But, on 24 December 2015, the city council eliminated the "congregate care facility" land use designation from municipal code. On 19 February 2016, the planning division issued an administrative interpretation determining that TOSA should be a "community correctional facility"[2] instead. But TOSA submitted an appeal not to be classified as a "community correctional facility", because this is an institutional designation that could not operate in the proposed site which was zoned as "moderate density multifamily residential".

On 5 July of 2016, TOSA submitted a revised request to operate as a "vocational and professional training education facility", but on 19 August 2016, the planning division determined that the TOSA proposal should be classified as a "large group home"[3]. On this notice, the city wrote that still, TOSA could not operate on the proposed location because their property was closer than 800 feet from Avenues Courtyard Assisted Senior Living Facility, another group home. TOSA prepared an appeal to notify the city that the 800 feet spacing requirement may violate the Fair Housing Act. The city lawyer determined that TOSA was right in their assessment, which allowed TOSA to operate in their current property. On 8 November 2016, the planning department issued an updated interpretation that confirmed the classification of TOSA as a "group home". They also informed TOSA about the requirement to submit a use permit.

A conditional use can only be rejected if the planning commission determines that potential harmful effects cannot be adequately addressed through the implementation of conditions. Some of the conditions that the planning division requested were for TOSA to register with the Utah Department of Human Services (DHS) to operate. The planning division suggested that TOSA should register using DHS's least restrictive license for group homes, known as a "recovery residence". Also, following crime prevention through environmental design (CPTED) strategies, lighting was required in all entrances, trees and shrubs needed to be trimmed regularly for visibility, and a fence needed to be installed on the property.

Five days before 22 March 2017, the city updated their interpretation, declaring that, under the group home designation, the business activities they were operating were not allowed—that is, a kitchen for their food trucks and parking for the food/moving truck vehicles. TOSA also stored and prepared food in their group kitchen to be sold at their food truck, and this constitutes a commercial activity. Because of this constraint, TOSA suggested that commercial use be permitted as an activity to their training. However, the staff concluded that while "vocational training" is commonly associated with group home use, commercial food preparation or storage is not typically connected to such a use. As a result, the planning division determined that the business kitchen activity could not be

authorized on the property either. To summarize, the staff report recommended that the planning commission required, as part of their conditional use, that no commercial activity should be permitted. TOSA contested that decision. They believed firmly that the group home definition was broad enough to incorporate their "rehabilitation" services (i.e., their vocational school). Based on that rationale, their "commercial activities" were not part of a business but were a part of their school. TOSA argued that parking their trucks and storing and preparing food for their food truck was the core function of their use as a group home.

"What we are trying to do is an odd duck. I think that is what has been difficult for the city and planning and zoning to figure out what to do with us. We do not fit into a box", said the chairman of the TOSA's board. He added,

> Any effort that tries to put it into a box that it doesn't belong in. . .literally kills it. Before opening this in Utah. I studied 20 other programs in a variety of states and in every place was treated as a rehab or a half-way house. As soon as they started to be treated that way and expected to comply with policies related to that, it starts changing the model to the point that it kills it. What you got is another rehab eventually.

A commissioner expressed his frustration with the planning staff recommendation of not allowing commercial uses,

> Here we have a model that clearly works, that is successful, that the community is resoundingly in favor for. It seems that we are throwing up some hurdles for them to cross that is disheartening and I can hear the frustration why we do not support something that works so well. I understand that these are administrative process, but to me, why are we not railing for this cause? Here we got a free rehab that prevents homelessness, it is in the East Side (wealthier part of town), no taxpayers money, it creates jobs, rehabilitates historic buildings, and it is saving lives. It is doing all the things we want!

Utah state law, however, says that there is no such thing as a use variance in Utah. TOSA is requesting that they be granted commercial uses (as interpreted by the planning staff) as a group home. Commercial uses are not typical of group homes and therefore what TOSA is trying to achieve does not fit in any city's zoning ordinance in the State of Utah. The city attorney explained to the planning commission, "What needs to happen for this use, as they are proposing to use it, is that we need to invent a new zone. It is more appropriate to change the zone, and presently we do not have this tool".

The commission agreed to starting such a petition. Still, until such a zoning ordinance is created through legislation, the commercial activities at TOSA should cease. The planning commission decided that they would approve the conditional use permit, allowing TOSA to operate as a group home, but that they would take out the restriction that said that commercial activities should cease. This decision was, after all, not a planning commission matter, but an interpretation of the zoning officer. Several planning commissioners sided with the applicant, who said that the staff report was contradictory because it allows vocational activities as an accessory use while prohibiting what was interpreted as commercial uses which were part of the group home vocational program. In other words, while the zoning administrator understood that TOSA could not have commercial vehicles on site, etc., the zoning administrator can also decide what the scope of the vocational use is. It was determined that TOSA still was within the 10-day period to submit an appeal to the hearing officer, and they could provide an interpretation of the interpretation. In the end, the hearing officer accepted their appeal based on the evidence shown—allowing TOSA to park their trucks and store and prepare food for their food truck as an accessory use (vocational training) of the group home.

## 6. Discussion

There are several lessons that can be learned by scholars and practitioners from this case study surrounding the strategies that might help in overcoming community opposition

to group homes which contributes to the existing literature [17,47]. For instance, the TOSA development team made a technically flawless and rational proposal, whose expert knowledge serves to favor the acceptability of the project. TOSA supporters, many of whom were clients, also made logical arguments about the value of TOSA that resulted in support of neighbors, the planning commission, and city council. Most importantly perhaps, TOSA engaged in public education, awareness-raising, and community engagement way before they went to the planning commission for approval.

They were able to do this through creative outreach, including the following approaches: (1) regularly attending community council meetings and creating an open forum for the community; (2) knocking on the doors of their neighbors and telling them about TOSA's mission and model; (3) sharing with people information about what they planned to do on the site; (4) listening to the needs and concerns of the community regarding site design and operations; (5) reinvesting and cleaning up the neighborhood; (6) increasing visibility on the street and public safety; (7) offering free community services as a way to be known and start developing trust. TOSA focuses on the strengths-based approach and promotes a culture of being a good neighbor and giving back to the community as "treatment" to achieve reintegration [6,33,71].

Models like TOSA, however, do not quite fit the land use and zoning regulations of many communities, which tend to practice exclusionary zoning [44,45,47]. As demonstrated by this case study, the process of determining that the proposal should go through a "conditional use permit" was a lengthy one. First, the city eliminated a whole designation from the municipal code's "congregate care facility" shortly after the TOSA application was filed, precisely because this application forced them to examine the land use compatibility concerns.

Second, the staff planner and the planning director determined that TOSA was a "community correctional facility"; with this designation, the city made an administrative decision that it would not allow TOSA to operate on the property they had purchased in a dense residential zone. After TOSA appealed, the city designated the property a "large group home". This also meant they could not operate in the current site because the code arbitrarily stated that group homes could not be within 800 feet of each other. After potential litigation under the Fair Housing Act, the city agreed with TOSA's request to be classified as a group home.

Third, the planning division interpreted TOSA's activities to include commercial uses as illegal—because educational or vocational training under the current zoning did not allow for dormitories. Together, these three zoning interpretation decisions undermined TOSA's application. At every step of the way, TOSA had to appeal and put a lot of effort into figuring out how to make their proposal work for the property they had already purchased. Ultimately, the decision comes down to the commissioners and to the hearing officer—who were key to changing interpretations among planning staff.

TOSA served to provoke a public debate over the relationship between law, zoning, and group homes in Salt Lake City. Practitioners can learn from this case study: if the administrative process fails to favor the public interest, then the legislative process should be petitioned by the planning commission to change the zoning laws that represent obstacles for the kind of development that the city wants to see [47]. As this case study has demonstrated, administrative processes can result in unjust outcomes when the case presented is an "odd duck", as the chairman of TOSA's board put it.

This case study shows how staff in the planning division use their power to interpret and analyze the zoning code in a way that was consistently used against TOSA. Planning officials seem to hide in the technicalities of the zoning code and ignore the intent of the ordinance. This case demonstrated that TOSA was able to mobilize neighbors in support and therefore match the power of the planning staff. Staff, however, tended to listen to the city lawyer interpretations—which, in this case, understood that the planners, by abiding to their interpretation, could be violating the Fair Housing Act. This case shows

that it is important to train and empower planning staff to best determine the intent of the ordinances.

This study offers insights and strategies for both researchers and practitioners. Based on the analysis of the case study, we can derive outcomes, highlight some lessons learned from the process and strategies for addressing community concerns related to group homes:

1. Start by engaging in continuous education and community involvement. It is crucial to initiate awareness campaigns. Actively participate in community meetings to foster discussions. Take the time to proactively communicate the purpose and model of the group home.
2. Implement outreach approaches that involve interactions with neighbors. Knock on doors and share project information. Genuinely listen to the needs and concerns of the community members. Additionally, consider offering services that benefit the community as a gesture of goodwill and establishing trust.
3. Adopt a strengths-based approach when communicating about the group homes' impacts on the community. Emphasize its role in promoting reintegration and highlight how it can contribute positively as a neighbor.
4. Thoroughly understand land use and zoning regulations specific to the area in question. Conduct research to ensure compliance with these regulations, even if they do not align perfectly with your proposed model—be prepared for modifications.
5. Be ready to advocate for changes in zoning laws if existing regulations pose obstacles against establishing group homes. Engage in petitioning processes directed towards planning commissions or legislative bodies, urging reconsideration or amendments that facilitate desired developments.
6. To increase the likelihood of overcoming resistance to group homes and fostering supportive environments, it is important to provide training and empowerment to planning staff members. They should be well-versed in interpreting zoning ordinances, with an emphasis on understanding the purpose rather than getting caught up in technicalities.

By implementing these practices, both communities and practitioners can create a welcoming and supportive atmosphere for group homes.

## 7. Conclusions

The decision of where and how to locate housing and services for formerly incarcerated individuals are land use and zoning decisions, which are made through law, precedence, and argumentation from stakeholders and decision makers. In this case study, surprisingly enough, TOSA encountered little NIMBY opposition, which has been discussed as a main concern in the previous planning literature [14,16,38,72]. Arguably, this is because they worked hard to introduce themselves to their neighbors. As other studies have shown, by having contact with TOSA students through community services as well as through their businesses, residents became more familiar with them; thus, negative attitudes towards the formerly incarcerated were reduced [8,73].

This case study contributes to others that show that, when issues are controversial, public education might go a long way to attain public support [8,39,49]. All of these findings support the idea that those with more information and exposure to reentry issues, as well as to the actors involved (e.g., planners, developers and clients), can play a role in increasing the acceptability of group homes and vocational programs [14,38]. This means that, to counteract NIMBY movements related to group homes and vocational programs, organizations that serve this population, planners, and others can influence public perception through education and advocacy.

Most often, homes for the formerly incarcerated need to go through a conditional use permit or zoning variance as well as meeting particular design and operation standards. Cities and communities, through their policies, can explicitly or implicitly support or not support organizations like TOSA. In many cities, land use regulations need to be changed to accommodate these kinds of operations. But the political will to approve conditional

uses or change ordinances still needs to be there. Much of the support or the lack thereof then comes back to decision makers, advocates, residents, and other stakeholders' will and their ability of argumentation, while navigating land use laws and regulations efficiently.

Scholars and planners can learn about how planning practice fails theory from this case study; often, planning staff interpret the zoning code by focusing on technical details and not paying that much attention to the intent of the legislation. Norm Krumholz and Paul Davidoff argue that planners should work from within to promote decision making that facilitates those that have been historically marginalized to achieve social justice goals and address historical socioeconomic inequities [11,12]. The mother who spoke about TOSA "saving lives" bring us to larger questions in planning research and practice as it relates to equity planning, which is taught in planning schools and argues that the goal of planning should be "to adopt policies, administrative practices, and resource allocations that expanded choices and opportunities for those who had the least" [13]. Given the growth of imprisonment, the limited resources available to finance housing, and issues of NIMBYs facing housing people who have been incarcerated, hopefully this case study has shown the value of housing assistance during reentry, why zoning obstacles should be overcome, as well as why academia and planning division should better train planners to achieve the intents of zoning ordinances [17,39].

In conclusion, it would be beneficial to include suggestions and supportive strategies for promoting change and addressing community opposition to group homes. These suggestions can serve as an agenda for policymakers and other important stakeholders in creating inclusive regulations and fostering successful models of reintegration. Here are a few additional recommendations:

1. Consider making amendments to zoning regulations that specifically cater to the needs of group homes. This may involve introducing articles or provisions that acknowledge the role of group homes in supporting vulnerable populations and facilitating community reintegration.

2. Look at density and proximity requirements for group homes, ensuring that they do not inadvertently hinder the establishment of facilities. Adopting approaches in these requirements like adjusting the distance between group homes or revisiting occupancy limits can help expand access to locations while respecting the essence of residential neighborhoods.

3. Introduce incentive programs aimed at encouraging community integration with group homes. These programs can offer other benefits to neighborhoods that actively embrace and support the presence of group homes fostering a culture of support.

4. Encouraging collaboration—it is important to promote planning approaches that involve stakeholders, including community members, group home operators, planning officials, and policymakers. By creating forums for discussion and decision making, collaborative planning can lead to well-informed and inclusive outcomes. This approach ensures that concerns are addressed and tailored solutions are developed for the benefit of both the community and group home residents.

5. Investing in infrastructure and services—it is crucial to invest in the development of infrastructure and services that can enhance the functionality and integration of group homes within communities. This includes facilities such as community centers, job training programs, mental health support services, and improved public transportation accessibility. Such investments contribute significantly to reintegration efforts and an improved quality of life for group home residents.

6. Sharing practices and networking—facilitating the sharing of practices and providing networking opportunities for stakeholders involved in group home development is essential. Conferences, workshops, and online platforms can serve as spaces where practitioners, researchers, and policymakers come together to exchange knowledge. By learning from case studies and collaborating on approaches to address community oppositions, we can promote the successful integration of group homes.

By incorporating these suggestions, this case study provides stakeholders with an agenda that actively promotes change. It encourages regulations while supporting reintegration models for group homes.

**Funding:** This research received no external funding.

**Data Availability Statement:** Data is unavailable due to privacy and ethical restrictions.

**Conflicts of Interest:** The author declares no conflicts of interests.

## Notes

1. Dwelling, Congregate Care Facility: A housing development designed, and managed to include facilities and common space that maximize the residents' potential for independent living [69].
2. Community Correctional Facility: means an institutional facility licensed by or contracted by the State of Utah to provide temporary occupancy for previously incarcerated persons or parole violators, which assists such persons in making a transition from a correctional institution environment to independent living [70].
3. Dwelling, Group Home (Large): A residential treatment facility, occupied by seven (7) or more individuals that provides a twenty four (24) hour group living environment for individuals unrelated to the owner or provider that offers room or board and specialized treatment.

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
