# Peer review of "The Value of Reentry Housing, Zoning, and “Not in My Back Yard” (NIMBY) Obstacles, and How to Overcome Them"

_land, doi:10.3390/land13030275_

Round 1

Reviewer 1 Report

Comments and Suggestions for Authors

The motivations and aims of this research are described clearly and completely. However, it is necessary for the authors to better highlight in the introductory section the limitations of the studies carried out on the impact of protection and urban planning policies on the relocation of previously incarcerated individuals.

In section 3 a more detailed and structured description of the data sources used is needed.

Furthermore, the tested process needs to be described more thoroughly in section 3.

In the final discussions, a proposal for best practices deduced from the experimentation on the case study must be summarized.

Comments on the Quality of English Language

Some grammar typos in the manuscript must be removed.

Author Response

Dear Reviewer,

Thank you for taking the time to review our manuscript. I truly appreciate your feedback and suggestions for improvement.

I completely agree with your suggestion to give emphasis to the limitations of studies regarding the effects of protection and urban planning policies on the relocation of previously incarcerated individuals in the introduction.

I carefully revise the manuscript to provide an understanding of the gaps in existing literature.

I also acknowledge your comment about the necessity for a well-structured explanation of our data sources in section 3. In the revised version I provide a description explaining their relevance to our research.

Furthermore, I adress your recommendation to provide an explanation of our tested process in section 3.

Lastly, I truly appreciate your suggestion to summarize recommendations based on the case study experimentation in the discussions.

Thank you for bringing the grammar typos in the manuscript to our attention. I used a proofreader to correct them to ensure that the final version of the manuscript is free from any language related issues. However, citations are verbatim, and I did not fix those.

Again, we sincerely appreciate your review and valuable suggestions.

Best regards,

Reviewer 2 Report

Comments and Suggestions for Authors

The article deals with an urgent and socially relevant topic, the re-entry of people released from incarceration, particularly focusing on dedicated housing which certainly deserve more attention.

The introduction provides a clear picture of the topic while explaining the scope of the paper.

In section 1, lines 55-57 the third point of motivations appears twice: merge them or include a fourth point.

Section 2 provides an exhaustive and complete context framework about the topic, the current gaps and the key challenges. However, it is not formally a literature review – which is a codified article typology where the choice of the keywords, the search engines, the sources are preliminarily discussed before the investigation itself is performed and the outcomes are critically analyzed according to time distribution, products typologies, topic clusters, etc. The section itself is well written and done, it is simply recommended to change the section title into “overview of the current state of the art in the literature” avoiding the confusion.

Section 3 is the weakest part of the article at the moment. This section should explain in a more detailed way the adopted methodology.

a)       the method section cannot simply be grounded on a single case study. The choice of the method must be justified and discussed against the current theories in the specific field. Thus, the theoretical assumptions, the overall method structure and the expected results must be introduced and described in advance, in any case before the case study – which is an applicative demonstration of the method – is introduced.

b)      It is highly recommended that a workflow diagram is included (if the author is not familiar with this diagram typology, the following website is suggested for instructions and guidance: https://www.smartdraw.com/workflow-diagram/). This tool will help to explain the process and how the different sources and tools are used to achieve the scope of the study.

c)       The method section must provide a clear picture of how the overall process works in order to let other researchers can replicate it elsewhere in another context.

A sub-section about the case study should follow.

The result section (section 4) requires introducing the outcomes in a clearer way:

a)       Which are the results? (what type of results are reported?) if they are mainly based on observations and quote of transcriptions this must be explained and, much more relevant, it must be explained how they are critically analyzed within the overall logic chain of the process (namely the method).

b)      An overall picture of the qualitative and quantitative results should be included.

c)       There is no need of a two-level substructure (section 4; 4.1; 4.1.1 just section 4 and 4.n if two main result typologies emerge).

There is a jump in the sub-section numbering: line 225 -> 4.1.1 and the following is on line 273 -> 4.3.2. Something is missing or the numbers are incorrect. By the way the titles are the same.

Section 5 should probably become a subsection of n. 4.

The discussion section is mainly focused on the case study while it should be addressed to the overall process. First: discuss the case study results against the expected outcomes, possibly highlighting what was in line with the method and what instead took a different direction (is this negatively affecting the process or is it an unexpected opportunity for further development?). Second: good pointing out the lesson learned but try to be more systematic in collecting them. Create clusters and use bullet points to list them in order to make the readers’ life easier in following.

More practical suggestions (e.g. amendments or new articles to be added in existing regulations) or supporting schemes or re-integration models should be included in the conclusion to actively promote a change offering policymakers and other key-players a to-do agenda.

Author Response

Comment: The article deals with an urgent and socially relevant topic, the re-entry of people released from incarceration, particularly focusing on dedicated housing which certainly deserve more attention.

The introduction provides a clear picture of the topic while explaining the scope of the paper.

In section 1, lines 55-57 the third point of motivations appears twice: merge them or include a fourth point.

Response: Thank you for pointing out that section 1 of the article contains duplicate information of the third point. The numbering was incorrect one should have been “third” and then “fourth.”

Comment: Section 2 provides an exhaustive and complete context framework about the topic, the current gaps, and the key challenges. However, it is not formally a literature review – which is a codified article typology where the choice of the keywords, the search engines, the sources are preliminarily discussed before the investigation itself is performed and the outcomes are critically analyzed according to time distribution, products typologies, topic clusters, etc. The section itself is well written and done, it is simply recommended to change the section title into “overview of the current state of the art in the literature” avoiding the confusion.

Response: Thank you for your input. I'm sorry if the section title caused any confusion. I will make the required adjustments. I changed the titled as suggested to better represent the content of that section.

Comment: Section 3 is the weakest part of the article at the moment. This section should explain in a more detailed way the adopted methodology. The method section cannot simply be grounded on a single case study. The choice of the method must be justified and discussed against the current theories in the specific field. Thus, the theoretical assumptions, the overall method structure and the expected results must be introduced and described in advance, in any case before the case study – which is an applicative demonstration of the method – is introduced. It is highly recommended that a workflow diagram is included (if the author is not familiar with this diagram typology, the following website is suggested for instructions and guidance: https://www.smartdraw.com/workflow-diagram/). This tool will help to explain the process and how the different sources and tools are used to achieve the scope of the study. The method section must provide a clear picture of how the overall process works to let other researchers can replicate it elsewhere in another context.

A sub-section about the case study should follow.

Response: Thank you for your feedback. To address this issue, I will revise the methodology section by including a description of the underlying theories, structure of the method, and expected outcomes. I incorporated a workflow diagram to visually demonstrate how different sources and tools are utilized. By the way thanks for sharing the tool. It took me a little to figure it out, but I love it and I think I am going to now use it all the time. Furthermore, I dedicated a subsection to the case study offering comprehensive information about the specific context.

Comment: The result section (section 4) requires introducing the outcomes in a clearer way:

Which are the results? (what type of results are reported?) if they are mainly based on observations and quote of transcriptions this must be explained and, much more relevant, it must be explained how they are critically analyzed within the overall logic chain of the process (namely the method). An overall picture of the qualitative and quantitative results should be included.

Response: Thank you for your feedback and suggestions. I made an introduction to the results section to provide an overview of the findings in this study. The results primarily consist of data obtained from quotes from transcriptions of comments, testimonies, and planning commission meetings. These qualitative sources were carefully analyzed using content analysis and thematic coding techniques as part of the research process. Through this analysis key themes and patterns related to challenges, barriers and the value of reentry housing projects were identified. The impact of zoning regulations and community opposition was also explored. Additionally, insights from documents, regulatory perspectives, media sources and stakeholder conversations were incorporated to enhance our understanding of the case study. In summary the results section presents a view that combines data and insights, from various sources to provide a well-rounded understanding of our findings.

Comment: There is no need of a two-level substructure (section 4; 4.1; 4.1.1 just section 4 and 4.n if two main result typologies emerge).

There is a jump in the sub-section numbering: line 225 -> 4.1.1 and the following is on line 273 -> 4.3.2. Something is missing or the numbers are incorrect. By the way the titles are the same.

Section 5 should probably become a subsection of n. 4.

Response: The numbering was heading were fixed.

Comment: The discussion section is mainly focused on the case study while it should be addressed to the overall process. First: discuss the case study results against the expected outcomes, possibly highlighting what was in line with the method and what instead took a different direction (is this negatively affecting the process or is it an unexpected opportunity for further development?). Second: good pointing out the lesson learned but try to be more systematic in collecting them. Create clusters and use bullet points to list them to make the readers’ life easier in following.

Response: Thank you for your feedback! I revised the discussion section to address the overall process and organize the lessons learned into clusters with bullet points for clarity.

Comment: More practical suggestions (e.g., amendments or new articles to be added in existing regulations) or supporting schemes or re-integration models should be included in the conclusion to actively promote a change offering policymakers and other key-players a to-do agenda.

Response: Thank you for your feedback. I provide recommendations for making changes to existing regulations. This will give policymakers and other individuals an agenda of what needs to be done to facilitate the establishment of group homes and support reintegration models.

Round 2

Reviewer 1 Report

Comments and Suggestions for Authors

The authors have responded fully and convincingly to all my comments. I recommend publishing this paper in the current version.

Reviewer 2 Report

Comments and Suggestions for Authors

The author integrated the manuscript to meet most of the raised issues, improving the overall quality and structure of the paper.

Comments on the Quality of English Language

Minor editing is needed to fix typos.